# Enabling targeted mass drug administration for schistosomiasis in north-western Tanzania: Exploring the use of geostatistical modeling to inform planning at sub-district level

Jake D. Mathewson[1,¤,‡,*], Linda van der Spek[1,‡], Humphrey D. Mazigo[2], George Kabona[3], Sake J. de Vlas[4], Andreas Nshala[5], Ente J. J. Rood[1,¤]

**1** Kit-Royal Tropical Institute, Epidemiology, Center for Applied Spatial Epidemiology (CASE), Amsterdam, The Netherlands, **2** School of Medicine, Department of Medical Parasitology & Entomology, Catholic University of Health and Allied Sciences, Mwanza, Tanzania, **3** Ministry of Health, National Neglected Tropical Diseases Control Programme, Dodoma, Tanzania, **4** Department of Public Health, Erasmus MC, University Medical Center Rotterdam, Rotterdam, The Netherlands, **5** Crown Agents, London United Kingdom

¤ Current address: Department of Epidemiology, KIT Royal Tropical Institute, Amsterdam, The Netherlands.
‡ These authors share first authorship on this work.
* j.mathewson@kit.nl

## Abstract

### Introduction

Schistosomiasis is a parasitic disease in Tanzania affecting over 50% of the population. Current control strategies involve mass drug administration (MDA) campaigns at the district level, which have led to problems of over- and under-treatment in different areas. WHO guidelines have called for more targeted MDA to circumvent these problems, however a scarcity of prevalence data inhibits decision makers from prioritizing sub-district areas for MDA. This study demonstrated how geostatistics can be used to inform planning for targeted MDA.

### Methods

Geostatistical sub-district (ward-level) prevalence estimates were generated through combining a zero-inflated poisson model and kriging approach (regression kriging). To make predictions, the model used prevalence survey data collected in 2021 of 17,400 school children in six regions of Tanzania, along with several open source ecological and socio-demographic variables with known associations with schistosomiasis.

### Results

The model results show that regression kriging can be used to effectively predict the ward level parasite prevalence of the two species of Schistosoma endemic to the study area. Kriging was found to further improve the regression model fit, with an adjusted R-squared

**Data Availability Statement:** Precision mapping data used in this study have been aggregated to various administrative levels and are publicly available for download on the ESPEN database. https://espen.afro.who.int/diseases/schistosomiasis Shapefiles, base layers for maps, and data for sociodemographic covariates are all open source and publicly available, and referenced within the document. The database of sociodemographic variables used for the ZIP model, in addition to variables calculated in QGIS like lake distance, has been additionally uploaded to Zenodo and shared via the following link: https://zenodo.org/records/10003824 Site based data from precision mapping can made be available with permission from NIMRI and the Republic of Tanzania. Permanent Secretary Ministry of Health Government City: Mtumba Afya road/Street P.O. Box 743 40478 DODOMA Alternatively, you may contact the Tanzania MoH, at ps@afya.go.tz.

**Funding:** The work for this publication was made possible through the funding of the ASCEND project from the foreign commonwealth and development office (FCDO). The reference number for the project is PO8374. https://www.gov.uk/government/organisations/foreign-commonwealth-development-office The funders had no role in study design, data collection and analysis, decision to publish, or preparation of the manuscript.

**Competing interests:** There are no known competing interests from any of the authors that would have served to bias the research or manuscript being submitted.

value of 0.51 and 0.32 for intestinal and urogenital schistosomiasis, respectively. Targeted treatment based on model predictions would represent a shift in treatment away from 193 wards estimated to be over-treated to 149 wards that would have been omitted from the district level MDA.

## Conclusions

Geostatistical models can help to support NTD program efficiency and reduce disease transmission by facilitating WHO recommended targeted MDA treatment through provision of prevalence estimates where data is scarce.

## Author summary

In Tanzania, schistosomiasis is a vast public health problem treated through mass drug administration (MDA) campaigns that target large groups of the population. Such mass drug administration (MDA) campaigns require significant amounts of resources challenging the capacity of chronically underfunded schistosomiasis control programs to sustain annually. The way in which MDA campaigns have been conducted, by targeting whole districts believed to be endemic, have not had optimal results for reducing disease transmission, have facilitated problems of under-treating in many population groups while over-treating other non-endemic areas, and have placed a significant strain on the limited resources available.

To circumvent such problems, the World Health Organization (WHO) has recommended that treatment efforts need to be more targeted to endemic communities and administrative areas within districts, and that MDA campaigns should be conducted at a sub-district level. The WHO also provides explicit guidelines for treatments of communities based on parasitological prevalence of schistosomiasis. While there is sound rationale in making this switch, most schistosomiasis endemic countries including Tanzania do not have adequate surveillance data to make such informed decisions at this level. Limited data on schistosomiasis prevalence inhibits disease control programs from making data informed decision on where to treat, as well as adhering to WHO recommendations for conducting targeted treatment at the sub-district level.

Geostatistical models are spatial analysis tools that have been used in the past to help predict the likelihood of disease prevalence in nearby areas where there is limited data. They are particularly helpful in predicting diseases like schistosomiasis that have strong associations with environmental and socio-demographic elements that we do have data on across much of the world. Publications on the use geostatistical models to predict schistosomiasis prevalence in different settings date back more than 20 years, but have not yet been integrated to aid the decision process of national programs, nor have they been specifically advocated for by the WHO. The study seeks to examine the implications of using model predictions to guide targeted, sub-district level treatment, in terms of populations requiring treatments when compared with conventional district level approaches.

This publication demonstrates that use of geostatistical models for predicting schistosomiasis is possible and can be a valuable tool to inform where to treat. It furthermore estimates that the populations eligible for targeted treatment are considerably different than the populations who would be receiving treatment under conventional district level

treatment approached, placing emphasis on the importance of switching to targeted treatment. The paper argues that uptake of geostatistical models can enable targeted planning below the district level, a major step to facilitate a systematic change that the WHO believes will drive down transmission of the disease, the importance of which cannot be understated in an area that is so historically burdened by the disease and its associated health complications.

## Introduction

The United Republic of Tanzania is the country with the world's second highest prevalence of schistosomiasis, a neglected tropical disease (NTD) of major public health concern caused by parasitic blood flukes [1–3]. In 2012, over 51% of the country's 43.5 million people were estimated to be infected with one of the two endemic schistosome species in Tanzania, *Schistosoma haematobium* and *Schistosoma mansoni* [4]. In spite of disease control efforts in the years since, schistosomiasis is still believed to be highly prevalent, particularly in the regions that make up the Lake Zone in the northwest part of the country [5]. While limited surveillance capacity and the nonspecific and often sub-clinical manifestations of the disease contribute to the difficulty of estimating schistosomiasis prevalence and morbidity rates [6], it is estimated to result in 1.5–2.5 million disability adjusted life years per year globally[7].

Transmission of schistosomiasis is strongly driven by socio-ecological factors, as it mainly occurs when humans interact with freshwater bodies that contain snail intermediate hosts and are contaminated by excreta of infected people [8,9]. The main control strategy recommended by the World Health Organization (WHO) is population wide preventive chemotherapy (PC) with praziquantel, which is delivered through Mass Drug Administration (MDA) campaigns. For MDA to be sustainable and effective, it should be supplemented by health education and safe water and sanitation interventions (WASH), and environmental management and snail control should be given to prevent re-emergence of the parasite in low prevalence settings [10]. In Tanzania, PC is often delivered in school based de-worming programs largely targeted at school-age children (SAC), since this group is relatively easy to reach and is expected to carry the highest burden of disease compared to other populations at high risk [11,12].

In Tanzania, these control strategies have yielded limited success in reaching disease control targets [13]. Of the 16 million individuals requiring preventive chemotherapy for schistosomiasis in 2020, only 3.2 million individuals actually received it [14]. Elimination has not been achieved in any of the endemic districts, and substantial variation in the community-level compliance with MDA treatments have been documented [15,16]. Limited disease prevalence data, sub-optimal surveillance systems and sparse resources for MDA campaigns make it challenging for disease control programs to target their efforts to the populations most in need [11]. While millions of people in need of PC in Tanzania will not receive it each year, district wide treatment campaigns have facilitated a practice of overmedicating some communities with low or no disease burden, which strains limited resources and increases the likelihood of adverse drug events[17–20]. The uncontrolled transmission of schistosomiasis and need for recurrent MDA campaigns at a district wide level has created a substantial economic and human resource burden on schistosomiasis control programs and increased dependence on foreign funding to sustain them [13,21].

Understanding of the geographic distribution of the disease is vital to enable more focused interventions and efficient use of resources [1,13,15,21]. In Tanzania and other high burden settings, schistosomiasis prevalence has been estimated by sampling a small selection of schools to

determine the endemicity and MDA eligibility of an entire district [22]. As schistosomiasis has been demonstrated to be heterogeneously distributed throughout populations in endemic districts, questions remain for the effectiveness and efficiency conducting MDA in an entire district that may contain hundreds of thousands of people [23]. District wide treatment has further been implicated in facilitating the aforementioned problems of over and under-treatment of certain populations [21]. Therefore, in 2021 in line with WHO recommendations, the government of Tanzania committed to shift from implementing MDA treatment per district-unit to the much smaller ward-unit, which would be enabled by mapping schistosomiasis at a finer geographical resolution, referred to as precision mapping [21,24,25]. Despite efforts and progress in precision mapping in the Lake Zone of Tanzania, the majority of wards still lack reliable prevalence data [5].

Since routine precision mapping of disease prevalence in each ward would be highly costly and not likely feasible at a large scale, the WHO has recommended the development of new cost effective tools to facilitate the process [25]. The use of geostatistical and spatial modeling has generated interest for its capacity to evaluate and map areas that may be endemic for schistosomiasis where there is limited surveillance or prevalence data [26–28], and has even been demonstrated to be a viable and cost-effective method of mapping schistosomiasis prevalence in certain settings (29). Regression techniques have additionally been used to estimate relationships between survey-based schistosomiasis prevalence and socio-ecological factors, and to predict prevalence at the un-sampled locations [30,31]. Some of these ecological factors, which have been shown to associate with the presence of schistosomiasis, are freely accessible and fairly well documented, and include distance to water bodies or wetlands, temperature, rainfalls, and altitude [32]. Elements that are less frequently incorporated in predictive models but have been well-established to be associated with schistosomiasis prevalence include WASH conditions and childhood malnutrition [33–36].

Although its coverage is still limited to a small proportion of wards, precision mapping data is becoming increasingly available in endemic settings [5,37]. Though the combination of precision mapping and geostatistical modeling techniques have been utilized to enable the low-cost and reproducible estimation of the geographical distribution of schistosomiasis prevalence, there is still limited evidence for the uptake of these techniques at scale to guide intervention strategies for disease control [25]. While previous spatial modelling studies have made estimations of prevalence at national and continental scales, there are few modeling studies that have offered reliable predictions of schistosomiasis prevalence at small enough geographical scale to facilitate the more targeted use of PC treatment at ward level [38].

The objective of this study is to evaluate how geostatistical modeling techniques can best be used to predict schistosomiasis prevalence at the ward level in Lake Zone, Tanzania. This is done by presenting an analytical study that uses precision mapping survey data and ecological covariates from open source data repositories to build a geostatistical model that creates prevalence estimates for each both *S. mansoni* and S. *haematobium* in wards of Lake Zone, Tanzania. The study seeks to examine the implications of using model predictions to guide targeted, ward level treatment, in terms of populations requiring treatments when compared with conventional district level approaches. Finally, the study attempts to demonstrate how data-driven targeted MDA at the ward level can be enabled using geostatistical models to bridge data availability gaps between wards with existing precision mapping data and those without.

## Methods

### Ethical considerations

For data collection procedures, IRB (Institutional Review Board) approval was given by the National Ethical Committee, National Institute for Medical Research Tanzania (NIMR/HQ/

R.8a/Vol. IX/3481), as well as by the concerning regional and district administrative authorities, and is available upon request. The study was conducted in accordance with the Declaration of Helsinki and International Conference on Harmonization Guideline on Good Clinical Practice (ICH-GCP). Formal written consent forms were signed by all parents or guardians whose children participated in the study. Children were furthermore only included in the precision mapping study if they had additionally agreed to the assent forms detailing the study methods and their right to withdraw from the study at any time, as explained in a previous publication on precision mapping in the Lake Zone[5]. Study population and sampling procedures.

In May 2021 schistosomiasis parasitemia assessments (precision mapping) were conducted through school surveys in six schistosomiasis endemic regions (Mwanza, Shinyanga, Mara, Simiyu, Kagera, and Kigoma) in the Lake Zone of Tanzania, surrounding Lake Victoria [5]. An overall number of 290 primary schools were surveyed, evenly distributed among the 29 districts and covering 223 (27%) of the 829 wards in the study area. In accordance with WHO guidelines, the survey sites were purposefully selected from three different ecological zones, based on proximity to water bodies [39]. To assess whether the 290 schools gave a good representation of the overall study area, main descriptive statistics were reported, and compared between sampled and non-sampled locations.

At each school, 30 boys and 30 girls aged 9–13 years were surveyed, adding up to 17,400 children. Urine filtration was used as the diagnostic test for urogenital schistosomiasis, with each sample examined by two technicians. The Kato Katz technique was used for intestinal schistosomiasis, with processed samples prepared on four slides (two sets of duplicate slides) using thick smear for two technicians to evaluate. Thick smears for each of the four slides were prepared using distinct areas of the stool sample to account for a potentially heterogeneous distribution of eggs within fecal samples. For quality assurance of both urine filtration and Kato Katz tests, 20% of all positive and negative samples were re-examined by a third technician blinded to the findings of the previous two technicians that had examined them. The information on the process of collection of samples is described in greater detail in a previous publication on the precision mapping survey [5]. Handheld global positioning system (GPS) devices were used to determine the geographic coordinates of each sample site.

**Data management and covariate selection.** Covariates were selected to be included in the models based on previously reported associations with schistosomiasis in past studies. A full list of covariates and their resolution and time frame is shown in Table 1 [26,32,40,41]. The density of snail species that can act as an intermediate host have been found to be affected by altitude, thriving in lower altitudes closer to sea level [32]. These snail species survival and fecundity have furthermore been shown to be influenced by ranges of land surface temperatures [32,42]. Terrain ruggedness was used as a proxy to estimate the likelihood of accumulation of standing water correlated with snail occurrence [40,43]. In past precision mapping of schistosomiasis, vegetation indices have also been used to predict areas endemic for schistosomiasis [26,44]. The distance to a lake, or large freshwater body, has been found to be a strong determinant for human water-contact behavior, thereby influencing the probability of schistosomiasis prevalence and intensity [26,44]. Furthermore, data on practice of open defecation (% of population), use of Improved Water Sources (IWS, % of population) and stunting rates (% of children under five) was retrieved from the 2015 Demographic and Health Survey (DHS) data repository and used as proxies for WASH and socio-economic conditions, which have been reported to be associated with schistosomiasis transmission at the community level [26,34,35]. All covariates were derived from open source data repositories and extracted as raster data covering the study area. Data analyzed in QGIS software were presented on open

**Table 1. Description of covariate data included in the model.**

| Variable | Resolution | Type | Time frame | Data Source |
|---|---|---|---|---|
| Spatial delineation of wards | NA | Vector (shapefile) | April 2018 | www.gadm.org, version 3.4 |
| Elevation (DEM, m) | 30 x 30 m | Raster grid | 2011 | ASTER GDEM V3 |
| Improved water source (% of pop.) | 5 x 5 km | Raster grid | 2015 | DHS Spatial Data Repository |
| Lake distance (m) | 1 x 1 m | Raster grid | - | Calculated in QGIS using OSM |
| Open defecation (% of pop.) | 5 x 5 km | Raster grid | 2015 | DHS Spatial Data Repository |
| Pop. density (pop./km$^2$)* | 100 x 100 m | Raster grid | 2020 | WorldPop Census data |
| Ruggedness (TRI, m) | 30 x 30 m | Raster grid | - | Calculated from DEM |
| Stunting (% of pop. <5y/o) | 5 x 5 km | Raster grid | 2015 | DHS Spatial Data Repository |
| Temperature (LST, ˚C) | 1 x 1 km | Raster grid | Avg. day temp. 2022 | Worldclim (v.2.1) |
| Vegetation (NDVI) | 250 x 250 m | Raster grid | Daytime avg. 16 days 2020 | MODIS v6.1 |
| Water index (NDWI) | 100 x 100 m | Raster grid | Avg. 2020 | MODIS v6.1 MOD13Q1 |
| Wetness (TWI) | 30 x 30 m | Raster grid | NA | Derived from GDEM |

DEM: Digital Elevation Model; TRI: Terrain Ruggedness Index; LST: Land Surface Temperature; NDVI: Normalized Difference Vegetation Index; NDWI: Normalized Difference Water Index; TWI: Topographic Wetness Index; OSM: Online Street Map. *Matched with UNPD 2020 estimates.

source base layers from the United States Geological Survey (USGS) to provide context of proximity to lakes [45].

## Statistical analysis

To allow for ward level comparisons, all ecological data at lower resolutions, shown in Table 1, were aggregated up to the level of the ward by taking the mean of all observations within a single ward. To assess whether our survey data provides a representative sample of the ecological conditions across the wards of the whole study area, two-sample t-tests were performed to confirm that the mean values of covariates from surveyed and non-surveyed wards did not differ significantly from each other. Zero-Inflated Poisson (ZIP) regression was used to account for an excess of zero prevalence counts which were observed in the survey data. ZIP models were used to explore associations between the different socio-ecological covariates and schistosomiasis prevalence. These models are specified by simultaneously estimating the probability of finding zero cases (logistic distribution) as well as the expected count of cases [46]. To assess whether different endemic schistosoma species (S. *haematobium* and *S. mansoni*) occurred in different environmental conditions, separate models were fitted for urogenital and intestinal schistosomiasis prevalence data. A backward stepwise selection method was applied to the initial models including all covariates as both predictors of the counts (Poisson) and as predictors of the excess zeros (logit). Model fit was assessed using the Akaike Information Criterion (AIC) [47]. The best fitting model was identified as the model with the lowest AIC score, indicating best model fit after stepwise removal covariates with the lowest partial fit. Covariates were only excluded if there were high degrees of collinearity or if they reduced the goodness of fit based on the AIC, not according to levels of significance in univariate analysis. The Vuong test, comparing the ZIP models with standard Poisson models, indicated significantly better fit of the ZIP model if $p < 0.05$ [48]. These procedures were performed using Stata (Stata 15SE) [49].

After model fitting, the model residuals were used to test for spatial dependence using the Moran's I statistic. This showed that the ZIP models were not effective in fully explaining the observed spatial clustering of schistosomiasis across schools. To improve the predictive accuracy of the model, Ordinary Kriging was applied to interpolate the regression residuals and add these to the ZIP model predictions [50,51]. This technique combines regression modelling

with geostatistical interpolation to optimize model fit in order to predict in locations where no data was collected while accounting for spatial dependencies in the observed data, referred throughout this paper as regression kriging (RK) [50]. Spherical semi-variograms were calculated based on ward centroids and used to assess the presence of spatial dependencies in model residuals over incremental distance. This was used to predict where ZIP models over or under predicted schistosomiasis prevalence. Tenfold cross-validation of the ZIP and RK models was performed, and the resulting adjusted $R^2$ indicated the proportion of variation caused by the independent variables. Regression kriging and cross-validation were performed using R Software [52–56].

## Calculating treatment targets

Wards were identified as in need of treatment when the predicted schistosomiasis parasitological prevalence of SAC was at least 10%, in line with WHO treatment recommendations [10]. The total population in need of PC treatment for intestinal and/or urogenital schistosomiasis was calculated by adding up the total school-aged population for all wards in need of treatment under either a district-level or a ward-level approach. The population of wards was calculated through zonal statistics in QGIS based on secondary demographic data, namely the pixel-level population density [57]. The number SAC per ward was calculated multiplying the ward level population and proportion of SAC per district based on secondary demographic data from the national bureau of statistics of Tanzania [58]. The number of required annual treatments was built around the assumptions that (1) there would be one round MDA per ward or district annually, which may be subject to change based on intensity of infection, and (2) only SAC would receive treatment, which may additionally change in the coming years as disease targets encompass more members of the community [10].

# Results

## Descriptive statistics

The characteristics of the population studied are displayed in Table 2. Precision mapping demonstrated that in the regions of Kigoma, Mara and Mwanza, each directly situated along either Lake Victoria or Lake Tanganyika, intestinal schistosomiasis was dominant, as shown in Fig 1 and S1 Table. Conversely, in Shinyanga and Simiyu regions, which both do not border a lake, urogenital schistosomiasis was much more commonly found than intestinal schistosomiasis.

**Table 2. Characteristics of children in the sample (n = 17,398).** Intensity of infection for urogenital schistosomiasis was determined through counting eggs per 10 ml of urine through urine filtration test. Intensity of infection for intestinal schistosomiasis was determined through counting the number of eggs in 1 gram of fecal samples through the Kato Katz technique.

| Characteristic | | No. | % |
|---|---|---|---|
| **Total Sampled Mean Ag** | | 17,398<br>11.05 | 100<br>- |
| **Male sex** | | 8,666 | 49.8 |
| **Urogenital SCH, % of sample (95% CI)** | | 969 | 5.6 |
| | Light infection (<50 eggs/10ml), % of cases | 731 | 75.4 |
| | Heavy infection (>50 eggs/10ml), % of cases | 238 | 24.6 |
| **Intestinal SCH, % of sample (95% CI)** | | 1,694 | 9.8 |
| | Light infection (<100 eggs/g) | 600 | 35.4 |
| | Moderate infection (100–400 eggs/g) | 693 | 40.9 |
| | Heavy infection (>400 eggs/g) | 401 | 23.7 |

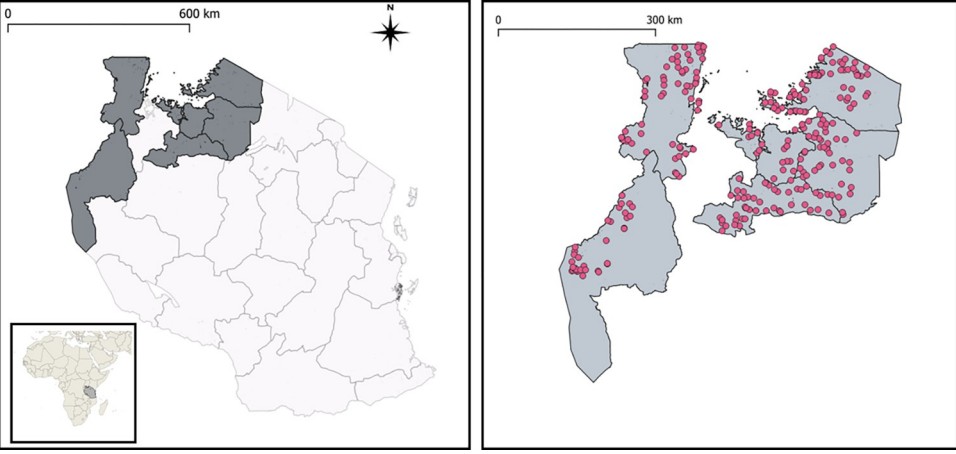

**Fig 1. The six highlighted Lake Zone regions examined in this study are presented on the left, and the distribution of the 290 sampled schools among those regions on the right.** Map was created in QGIS 3.30.1 using open source spatial data from GADM.org.

Kagera region carried a substantial burden of both urogenital (further inland) and intestinal schistosomiasis (along Lake Victoria). Aggregated to ward-level, intestinal and urogenital schistosomiasis were found in 109 and 117 of the 223 wards surveyed, respectively. In 42 wards both variants were present, while no schistosomiasis was found in just 41 of the surveyed wards. S2 and S3 Tables provide further insight into the socio-ecological statistics of sampled and non-sampled wards. Fig 1 gives a geographical overview of the 290 schools surveyed during precision mapping, and Fig 2 shows the proportion of children found in the survey to have schistosomiasis by strain and region.

## Non-spatial ZIP prediction model

Due to the high degree of collinearity, wetness (variance inflation factor (VIF) = 53.4; $R^2$ = 0.98) and temperature (VIF = 11.78; $R^2$ = 0.91) were excluded from the ZIP model. Table 3 shows the results of the ZIP model for covariates included in the best fitting models based on the lowest AIC. Increased prevalence of intestinal schistosomiasis was associated with areas of higher population density and stunting rates. Areas with decreased surveyed prevalence of

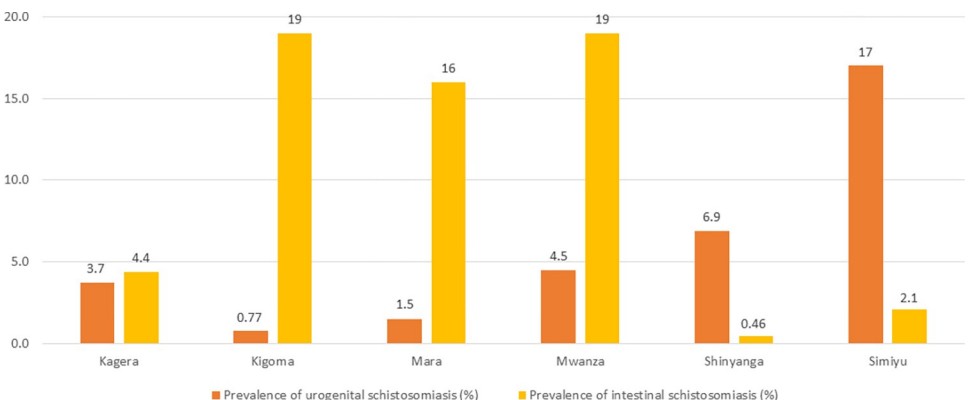

**Fig 2. Prevalence of schistosomiasis in Lake Zone regions.**

**Table 3. Zero-inflated Poisson model (ZIP) for intestinal and urogenital schistosomiasis at ward level.**

| | Intestinal schistosomiasis | | | Urogenital schistosomiasis | | |
|---|---|---|---|---|---|---|
| | Est. | P>z | 95% CI | Est. | P>z | 95% CI |
| **Count model** | | | | | | |
| (Intercept) | 0.43 | 0.04 | 0.01, 0.85 | -3.18 | 0.00 | -4.16, -2.21 |
| Elevation (DEM, m) | | | | 1.71 | 0.00 | 0.90, 2.53 |
| Lake distance (m) | -0.003 | 0.00 | -0.003, -0.002 | 0.001 | 0.00 | 0.000, 0.001 |
| Open defecation (% of pop.) | | | | -3.66 | 0.00 | -5.28, -2.04 |
| Population density (pop./km$^2$) | 0.01 | 0.00 | 0.00, 0.01 | -0.01 | 0.01 | -0.02, 0.00 |
| Ruggedness (TRI, m) | -0.01 | 0.00 | -0.01, -0.01 | -0.08 | 0.00 | -0.10, -0.05 |
| Stunting (% of pop. <5 y/o) | 3.24 | 0.00 | 2.26, 4.21 | -3.14 | 0.00 | -4.93, -1.34 |
| Vegetation (NDVI) | -1.08 | 0.00 | -1.75, -0.41 | 4.03 | 0.00 | 2.37, 5.69 |
| Water index (NDWI) | 2.45 | 0.00 | 1.57, 3.32 | 2.20 | 0.16 | -0.84, 5.25 |
| Water source (% of pop.) | -1.47 | 0.00 | -1.71, -1.23 | -0.47 | 0.07 | -0.98, 0.03 |
| Wetness (TWI) | -3.14 | 0.00 | -3.68, -2.60 | | | |
| **Logit model** | | | | | | |
| (Intercept) | -6.45 | 0.01 | -11.59, -1.31 | -2.36 | 0.17 | -5.70, 0.98 |
| Elevation (DEM, m) | 0.004 | 0.09 | 0.00, 0.01 | | | |
| Lake distance (m) | 0.003 | 0.00 | 0.001, 0.004 | -0.002 | 0.01 | -0.003, 0.00 |
| Open defecation (% of pop.) | | | | -8.17 | 0.03 | -15.33, -1.00 |
| Population density (pop./km$^2$) | -0.12 | 0.00 | -0.19, -0.06 | | | |
| Ruggedness (TRI, m) | | | | 0.10 | 0.00 | 0.03, 0.17 |
| Stunting (% of pop. <5 y/o) | -9.02 | 0.01 | -16.12, -1.92 | | | |
| Vegetation (NDVI) | 11.64 | 0.00 | 3.90, 19.37 | -8.01 | 0.01 | -14.00, -2.02 |
| Water index (NDWI) | 11.68 | 0.06 | -0.45, 23.81 | -9.52 | 0.04 | -18.47, -0.56 |
| Water source (% of pop.) | | | | -3.10 | 0.00 | -5.18, -1.01 |
| Wetness (TWI) | | | | -1.66 | 0.20 | -4.19, 0.88 |

*The ZIP model is composed of a (1) Count model (Poisson): Log of the ratio of expected counts (incidence rate ratios) and a (2) Logit model: Log odds of zero prevalence. When testing for model fit, the Vuong test: p = 0.00 showed that the ZIP model outperformed a standard Poisson model. Tenfold cross-validated models: intestinal schistosomiasis Adj. R$^2$ = 0.40; urogenital schistosomiasis Adj. R$^2$ = 0.23.*

intestinal schistosomiasis were found to be associated with further distances from a lake, terrain ruggedness, vegetation, wetness, and use of an improved water source. Surveyed prevalence of urogenital schistosomiasis was associated with increased elevation, lake distance, vegetation, and water source improvement, while negatively affected by increased open defecation, population density, terrain ruggedness, and stunting. The tenfold cross-validated adjusted R$^2$ indicated that 40.0% and 23.3% of variance for intestinal and urogenital schistosomiasis, respectively, was explained by the models. In S1 and S2 Figs, ZIP-predicted prevalence of intestinal and urogenital schistosomiasis, were shown on the map.

## Geostatistical prediction model

The Moran's I statistics (intestinal = 0.193 and p = 0.001; urogenital = 0.107 and p = 0.003; S3–S8 Figs) showed that the ZIP models did not effectively explain the spatial patterns observed in ward prevalence data. Residuals of the ZIP model, calculated by the difference between model predictions and observed prevalence, were subsequently interpolated using ordinary kriging based on the semivariogram (S9 and S10 Figs). Figs 3 and 4 show the ultimate prevalence predictions after adding the kriging results to the ZIP model predictions. As reflected by the

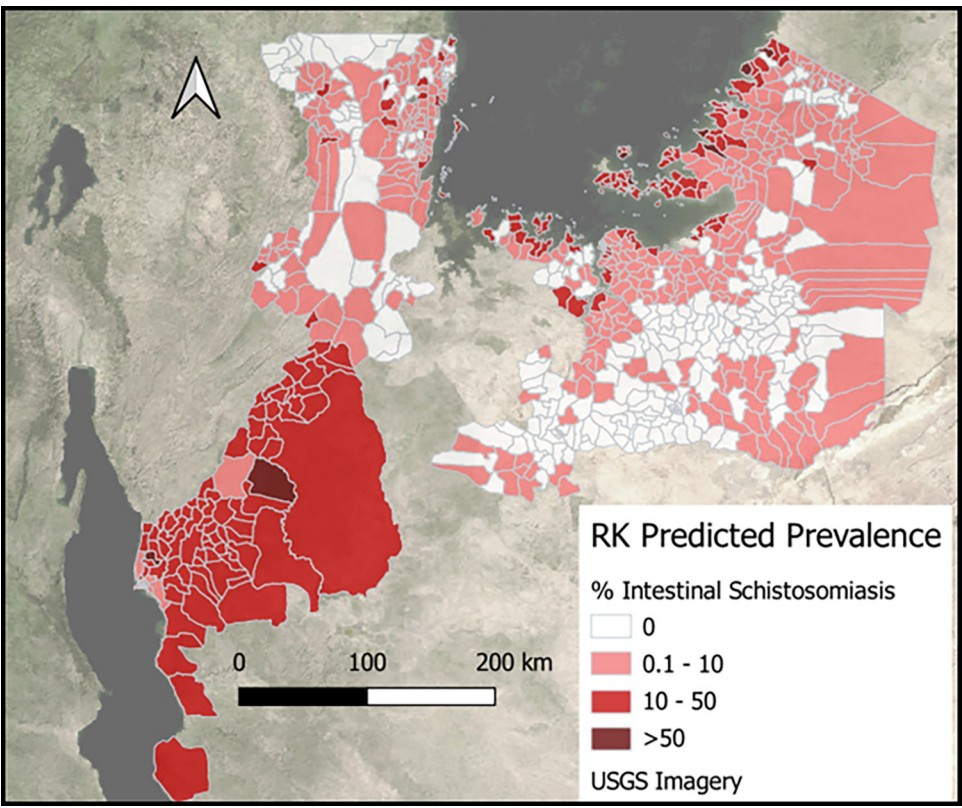

**Fig 3. Intestinal schistosomiasis prevalence predictions of the geospatial (regression kriging) model in wards in Lake Zone, Tanzania.** The model predicted high prevalence in many wards contiguous and proximal to lakes. Map was created in QGIS 3.30.1 using open source shapefiles and spatial data from GADM.org. Data were presented on open source base layers from the United States Geological Survey (USGS).

scatterplots in Fig 5, the adjusted $R^2$ of 0.51 for intestinal schistosomiasis and of 0.32 for urogenital schistosomiasis, kriging improved the model fit.

## Implications for MDA planning

Fig 6 provides an overview of the model-based treatment requirements at the ward level, as well as the districts that are treated when surveillance data is aggregated to the district level to inform MDA planning. Table 4 summarizes the treatment implications, showing that the spatial model-based approach identifies almost 2.1 million school-age children in need of treatment for urogenital or intestinal schistosomiasis, a number similar to the nearly 2.2 estimated school aged children using conventional district level approaches. However, the spatial model-based approach reduces the number of wards to be treated by 11% (47 wards), which are distributed throughout the map as shown in Fig 6. Fig 7 displays the wards predicted to be over- and under-treated by the conventional district-level approach. The model predicts that 951,000 SAC in 193 wards would be over-treated, and 929,000 SAC in 149 wards would be under-treated.

## Discussion

The study used geo-statistical modeling, specifically regression kriging, to estimate schistosomiasis prevalence in wards in Lake Zone, Tanzania. Results from the model demonstrate that

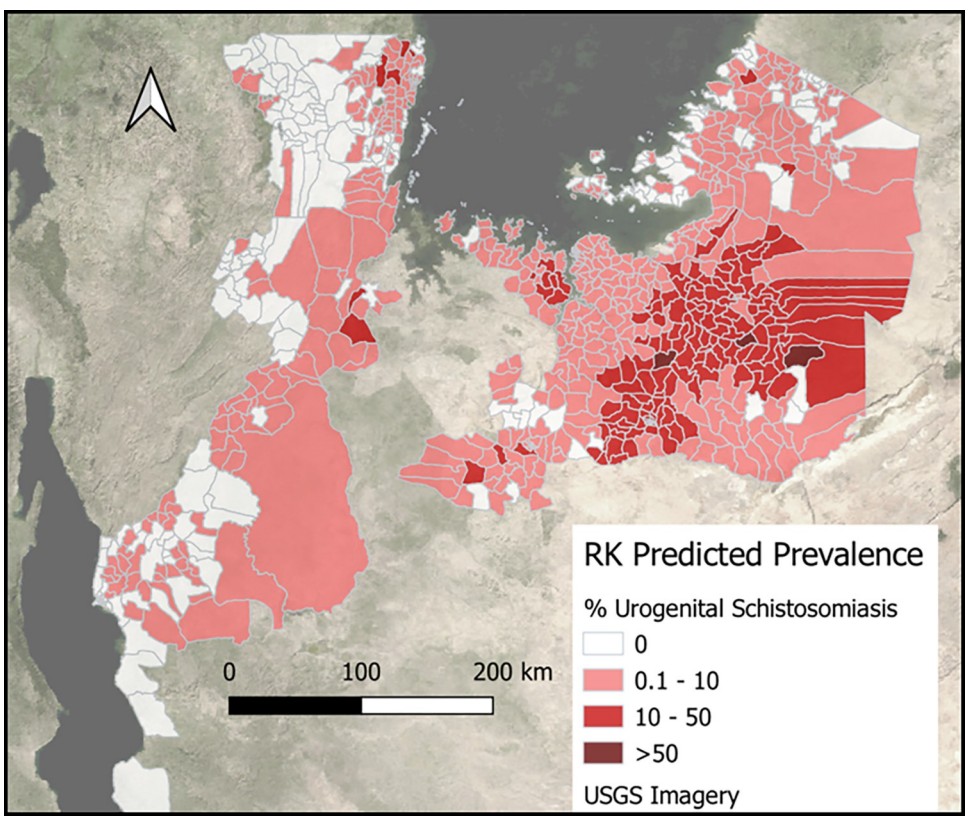

**Fig 4. Urogenital schistosomiasis prevalence predictions from the geospatial (regression kriging) model in wards in Lake Zone, Tanzania.** Urogenital schistosomiasis was predicted to be highly prevalent across the study region with wards having a high predicted prevalence estimated to be further inland than intestinal schistosomiasis. Map was created in QGIS 3.30.1 using open source shapefiles and spatial data from GADM.org. Data were presented on open source base layers from the United States Geological Survey (USGS).

use of a data driven approach to guide ward level MDA is feasible, and estimated to substantially reduce over-treatment (951 thousand SAC, 193 wards) and under-treatment (929 thousand SAC, 149 wards). Use of a ward level approach estimated an 11% reduction in wards that

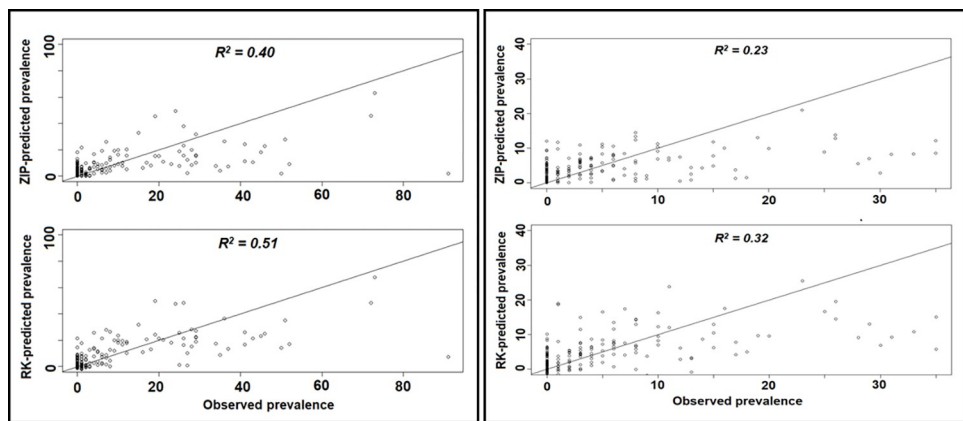

**Fig 5. Scatterplot to assess the tenfold cross-validated linear relationship between ZIP and RK predicted values and the observed precision mapping values, for intestinal (left) and urogenital (right) schistosomiasis.**

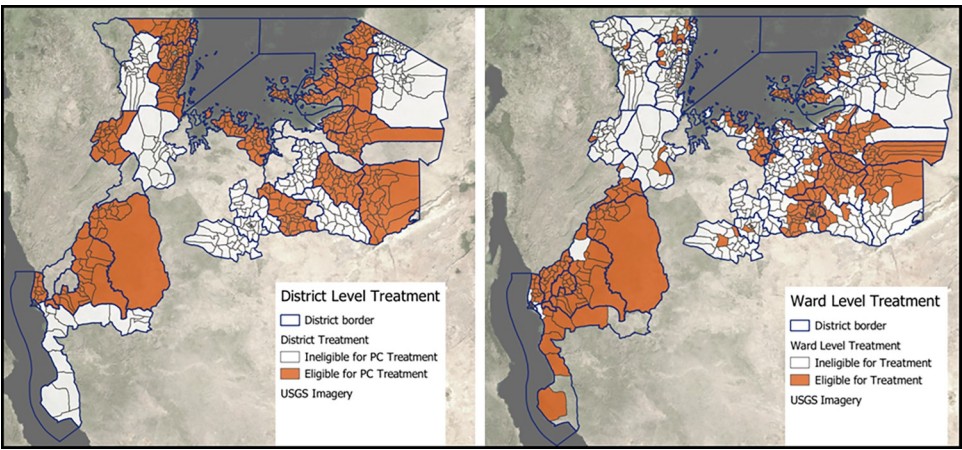

**Fig 6. Comparing the implications of PC eligibility at a conventional district-level approach based upon school survey results (left) with that of a ward level approach based on model estimates (right).** District level treatments needs were determined by a mean observed prevalence in school survey sites exceeding the 10% cutoff threshold, while ward level treatment needs were estimated using geostatistical model predictions to supplement observed prevalence from school surveys in wards where available. Map was created in QGIS 3.30.1 using open source shapefiles and spatial data from GADM.org. Data were presented on open source base layers from the United States Geological Survey (USGS).

needed to be treated compared to a conventional district level treatment approach, with the overall treatment numbers required 2.13 and 2.17 million SAC, respectively. Model predictions estimated intestinal schistosomiasis to be more commonly distributed in wards found in regions contiguous with Lake Victoria and Lake Tanganyika, while estimates for urogenital schistosomiasis were more commonly distributed in wards further inland. Results of the study further demonstrated that the methods of using a geospatial regression kriging approach to estimate schistosomiasis prevalence in wards was suitable over other methods tested, where the model using ZIP regression had a better fit than through use of Poisson regression, and where adding kriging further improved the model fit. The study also found that routine and publicly available data such as population density, stunting rate, distance to a lake, terrain ruggedness, vegetation, wetness, and use of improved water sources can be used to generate prevalence estimates of both types of schistosomiasis at a sub-district, ward level.

**Table 4. Implications for numbers of treatments needed from different methods of estimating prevalence.** The surveyed data model looks at the number of treatments needed when using conventional district level approach to MDA using results from the school surveys. ZIP and geospatial model estimate treatments using a more targeted ward level approach enabled by creating estimates for all wards.

| Model | Unit | Prevalence | Schistosomiasis | Wards | Population | SAC |
|---|---|---|---|---|---|---|
| **Surveyed data** | District | Mean observed | Urogenital | 155 | 2,965,000 | 877,000 |
| | | | Intestinal | 268 | 4,840,000 | 1,447,000 |
| | | | Total* | 423 | 7,249,000 | 2,172,000 |
| **Prediction model (ZIP)** | Ward | Predicted (ecological risk) | Urogenital | 123 | 1,909,000 | 579,000 |
| | | | Intestinal | 235 | 4,666,000 | 1,371,000 |
| | | | Total* | 357 | 6,563,000 | 1,946,000 |
| **Geostatistical model (RK)** | Ward | Predicted (ecological risk + observed prevalence) | Urogenital | 154 | 2,498,000 | 760,000 |
| | | | Intestinal | 227 | 4,744,000 | 1,398,000 |
| | | | Total* | 376 | 7,155,000 | 2,132,000 |

*Either urogenital or intestinal schistosomiasis to be treated for. Cut-off value for treatment eligibility: prevalence ≥ 10%. SAC: School-Age Children, 5–15 y/o. Population numbers were based on projected ward-level population density and regional age proportions [41].

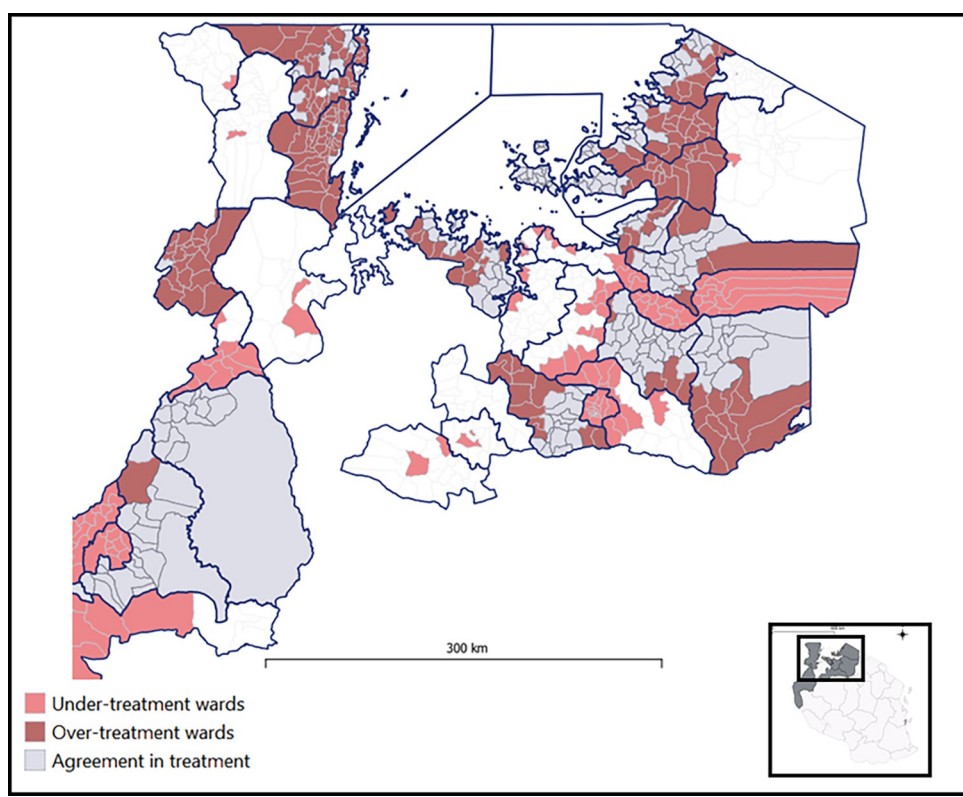

**Fig 7. Overview of treatment status of each ward based upon the model estimates.** The dark red wards represent wards were not estimated to require treatment, however would be treated under a district level approach. Therefore, they are labelled as "over-treatment" wards. The light red color shows wards that are predicted to be above the treatment threshold but would not be treated under a district level approach, and therefor "under-treatment wards". The gray wards would be treated in both model-based targeted treatment approach and the conventional district level approach, and non-shaded wards would not be treated in either. Map was created in QGIS 3.30.1 using open source shapefiles and spatial data from GADM.org.

A key object of this study is to investigate the potential for reducing over and under-treatment through a more targeted approach to MDA using modelled prevalence estimates. While the overall number of treatments necessary to treat against either species of schistosomiasis were similar in both ward level and district level approaches, the number of wards, and subsequently SAC, estimated to be over or under-treated were substantial. Figs 6 and 7 highlight areas in which the model estimated wards that needed PC treatment where a district level approach would not have provided it, resulting from the limited areas in which it is able to hold precision mapping. Treating in endemic wards where MDA was not previously occurring due to a lack of data would have a large health impact on populations affected, reducing local transmission and likelihood of associated morbidities. Additionally, avoiding the over treatment of wards helps control programs to conserve limited resources on costly MDA campaigns while averting potential adverse drug events in communities that don't need require treatment, and better preserve trust between drug distribution programs and at risk populations. Regardless of the error associated with the precision of the model, the granular outputs of ward level predictions allows for a critical appraisal of which wards to treat in the region that has not previously existed for the Lake Zone.

Consistency between the geographic distribution patterns of model predictions for intestinal and urogenital schistosomiasis prevalence with past research findings increased confidence

in the model outputs, and subsequently, the study approach [29,59–61]. This study added to an existing body of research demonstrating that the use of geostatistical modeling, specifically using regression kriging, can be a suitable way of predicting the spatial pre-control distribution of schistosomiasis [62,63]. Kriging can furthermore help for correcting for superfluous data patterns that cannot be explained with existing data, thus normalizing predictions to make them more accurate. This normalization, however, may also skew predictions closer to the mean in areas of high burden, as seen in Fig 5 where no geostatistical model estimates were as high as some of the observed wards. The comparison of different modeling approaches, including Poisson regression and zero-inflated Poisson (ZIP) regression, showed regression kriging to be the best of the examined methods for predicting schistosomiasis prevalence in the study area. This further supports the use of geostatistical models and regression kriging in future studies to add to existing surveillance data to improve upon the prediction of schistosomiasis prevalence. The relatively large sample size of the study, which included over 17,400 school-aged children from 223 wards, provided further confidence in the reliability of the model predictions. The increasing use of precision mapping in Tanzania and other settings will provide opportunities to replicate and build upon this study to create useful schistosomiasis prevalence predictions at the sub district level [5,37].

The ability to predict schistosomiasis prevalence at ward level could have important policy implications for the control and management of schistosomiasis in a number of settings. As the WHO now advocates for more targeted delivery of MDA at a sub-district level in spite of the lack of available data that can guide such decision, it is a key moment to start integrating tools that could facilitate control programs in more informed planning on where to allocate limited resources for disease control [25]. Since the study shows that use of geostatistical models can provide a data-driven approach to guide ward-level MDA that is both feasible and estimated to substantially reduce over- and under-treatment, public health officials and policymakers should investigate whether they can adopt such approaches in their respective settings. Central disease control agencies and donors should further investigate the suitability of this approach and provide guidance to facilitate national programs in making this large strategic shift to more targeted MDA.

We recommend using this tool as an addition to existing precision mapping data to guide targeted MDA in a pilot program where the precision of model predictions can further be evaluated. We offer these tools as a means to supplement expanded precision mapping efforts, and in no way advocate for use of modeling as a replacement. Precision mapping could be executed in a more cost effective way if geostatistics were accounted for during survey design, such as by using a lattice and close pairs design, to systematically map disease burden in a larger area of the country using fewer resources [29]. Model predictions should be used as a guiding component in the decision making process in addition to other relevant data, like past reported PC coverage, and cross sectoral collaborations with vector control and WASH interventions. Given the non-static nature of schistosomiasis prevalence over time in a given setting, models should be updated regularly with recent data, and preferably used on a real time monitoring platform where other relevant data sources can be viewed. Finally, we recommend further investigation and continued improvement of regression kriging and geostatistical modeling methods through future studies, as well as field evaluations of model performance. We recommend subsequent evaluations to also investigate costing to better understand the financial benefits or disadvantage of using this approach.

A limitation of this study was not accounting for past PC reported drug coverage in the model. While past drug coverage is both relevant and has been demonstrated to be associated with schistosomiasis prevalence, reported coverage was left out of the model because reported drug coverage data from MDA campaigns have been shown to be highly variable in accuracy,

often incorrectly describing the proportion of the target population that received the drug [64–67]. Furthermore, this coverage data is reported at the district level, which even when accurate, may not homogenously describe the wards within the districts. As the purpose of the model was to be able to enable targeted coverage based upon ecological and socio-demographic risk for schistosomiasis infection, the researchers strived to use covariates with data sources available at or below the ward level. Finally, none of the districts within the Lake Zone regions in the study area were found to have completed the 5 rounds of effective MDA that would provide an indication for not recommending further MDA campaigns per WHO guidelines [10,68].

In subsequent modeling, it will be important to find a way to account for past PC coverage, particularly when sub-district MDA starts to become more commonly used in Tanzania. Further limitations of the study include the extraction and aggregation of data from different sources with varying timeframes, scales and methods, which may have reduced the accuracy of some of the predictions [31]. Additionally, the modeling approach used may have led to an underestimation of standard errors in the ZIP model, however, since the model did not exclude covariates based on their univariate associations, but rather the overall predictive accuracy of the model, it is unlikely this would have altered the variable composition of the geostatistical models. Because of this, care should be taken in interpretation of associations of individual covariates with schistosomiasis prevalence. The use of proxy variables and other variables indirectly associated with schistosomiasis prevalence may have decreased validity and increased the chance of errors. The schools being sampled in a non-random and not fully transparent way poses a risk to validity as well, though researchers have stated that random selection of schools should be avoided due to the focal geographical distribution of schistosomiasis [11]. Future studies can be enriched with additional information from surveys (e.g., intensity of infection, co-infection with other NTDs, socio-economic status, age, and sex), improved census data, and past MDA coverage data at ward level.

In conclusion, the results from this study demonstrate that geostatistical predictive models, specifically using regression kriging, can be a viable tool to supplement the existing decision making process for schistosomiasis control programs to determine where to implement MDA in Lake Zone, Tanzania, as well as other schistosomiasis endemic settings where these methods can be replicated. These methods provide the capacity to guide interventions and facilitate more targeted PC treatment at ward level. This in turn can mean a reduction of over and under-treatment, particularly when guiding interventions to areas not previously receiving treatment, something that can significantly improve the efficiency of disease control programs and the health outcomes of communities that benefit from improved drug coverage. In addition to guiding the selection of MDA locations, prevalence predictions can be used to calculate number of treatments needed and advise on where to conduct future precision mapping efforts. In Tanzania, use of model outputs can help to circumvent the challenges of scarce ward level prevalence data, limited surveillance capacity, and lack of feasibility in attempting comprehensive precision mapping coverage at ward level. Going forward it will be important to evaluate how effectively these models can predict prevalence, how to improve upon predictive accuracy, and how to integrate the use of such models in existing systems so that they may be added to the decision making arsenal to better control schistosomiasis.

## Supporting information

**S1 Table. Prevalence of schistosomiasis per Lake Zone region.**
(XLSX)

**S2 Table. Descriptive statistics for socio-ecological variables across all wards.**
(XLSX)

**S3 Table. Comparative statistics for socio-ecological variables in sampled and unsampled wards.**
(XLSX)

**S1 Fig. Proportion of ZIP-predicted intestinal SCH per ward.**
(TIF)

**S2 Fig. Proportion of ZIP-predicted urogenital SCH per ward.**
(TIF)

**S3 Fig. Connectivity map and histogram.** A spatial weights file was generated iteratively to determine the contiguity structure for each ward (threshold: 53597.2m; inverse distance with power: 2; Euclidean distance; queen; order of contiguity: 1st; neighbors: max. 27; min. 1; median: 15).
(TIF)

**S4 Fig. Univariate Moran's I graphs.** Moran's I for the ZIP prediction residuals for intestinal (pseudo p-value of $p = 0.001$) and urogenital (pseudo p-value of $p = 0.003$) SCH.
(TIF)

**S5 Fig. Spatial correlogram of urogenital SCH residuals.** Max. distance = 350km. Zero auto-correlation at 105km. Frequency indicates the number of pairs (total: 19857).
(TIF)

**S6 Fig. Spatial correlogram of intestinal SCH residuals.** Max. distance = 680km. Zero auto-correlation at 141km. Frequency indicates the number of pairs (total: 24976).
(TIF)

**S7 Fig. Residuals of ZIP-predicted intestinal SCH in a box map.** Hinge = 1.5.
(TIF)

**S8 Fig. Residuals of ZIP-predicted urogenital SCH in a box map.** Hinge = 1.5.
(TIF)

**S9 Fig. Semivariogram of GLM residuals for intestinal SCH.** Spherical model with a partial sill of 75; a range of 50,000m; and a nugget of 10.
(TIF)

**S10 Fig. Semivariogram of GLM residuals for urogenital SCH.** Spherical model with a partial sill of 40; a range of 90,000m; and a nugget of 1.
(TIF)

## Acknowledgments

We would like to thank the entire team in Tanzania for their careful execution of the disease specific assessment precision mapping intervention which provided contextual data to make building this model possible. This includes the Tanzania NTDCP, CUHAS, as well members of the ASCEND program and all supportive implementing partners. We would like to specifically acknowledge Crown Agents for their supportive program management and Elodie Yard from Oriole Global Health for managing and coordinating field implementation. We would like to further acknowledge the community leaders, school administrators, and many parents

who help to make interventions for schistosomiasis like disease specific assessments possible in participating administrative units.

## Author Contributions

**Conceptualization:** Andreas Nshala, Ente J. J. Rood.

**Data curation:** Jake D. Mathewson, Linda van der Spek, Humphrey D. Mazigo, Ente J. J. Rood.

**Formal analysis:** Jake D. Mathewson, Linda van der Spek, Ente J. J. Rood.

**Funding acquisition:** Ente J. J. Rood.

**Investigation:** Jake D. Mathewson, Linda van der Spek, Humphrey D. Mazigo, George Kabona.

**Methodology:** Jake D. Mathewson, Linda van der Spek, Ente J. J. Rood.

**Project administration:** Jake D. Mathewson, Humphrey D. Mazigo, Andreas Nshala, Ente J. J. Rood.

**Resources:** Linda van der Spek.

**Software:** Ente J. J. Rood.

**Supervision:** Jake D. Mathewson, Humphrey D. Mazigo, George Kabona, Sake J. de Vlas, Ente J. J. Rood.

**Validation:** Jake D. Mathewson, Ente J. J. Rood.

**Visualization:** Jake D. Mathewson, Linda van der Spek, Ente J. J. Rood.

**Writing – original draft:** Jake D. Mathewson, Linda van der Spek.

**Writing – review & editing:** Jake D. Mathewson, Sake J. de Vlas, Ente J. J. Rood.

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
