## [Decision Letter · Decision Letter 0]

13 Jul 2023

Dear Mr Mathewson,

Thank you very much for submitting your manuscript "Enabling targeted mass drug administration for schistosomiasis in north-western Tanzania: Exploring the use of geostatistical modeling to inform planning at sub-district level" for consideration at PLOS Neglected Tropical Diseases. As with all papers reviewed by the journal, your manuscript was reviewed by members of the editorial board and by several independent reviewers. In light of the reviews (below this email), we would like to invite the resubmission of a significantly-revised version that takes into account the reviewers' comments. 

We cannot make any decision about publication until we have seen the revised manuscript and your response to the reviewers' comments. Your revised manuscript is also likely to be sent to reviewers for further evaluation.

Sincerely,

Philip T. LoVerde

Guest Editor

Eva Clark

Section Editor

Reviewer's Responses to Questions

**Key Review Criteria Required for Acceptance?**

**Methods**

-Are the objectives of the study clearly articulated with a clear testable hypothesis stated?

-Is the study design appropriate to address the stated objectives?

-Is the population clearly described and appropriate for the hypothesis being tested?

-Is the sample size sufficient to ensure adequate power to address the hypothesis being tested?

-Were correct statistical analysis used to support conclusions?

-Are there concerns about ethical or regulatory requirements being met?

Reviewer #1: - The authors state in the Discussion that not including past MDA coverage in the model is a limitation. However, they don't explain why this couldn't have been included, since surely this data would have been available from the schools that were surveyed (e.g., length of time since last MDA as a variable)?

- I am not a statistician, so cannot comment critically on the methods that were used.

Reviewer #2: All yes.

Reviewer #3: (No Response)

**Results**

-Does the analysis presented match the analysis plan?

-Are the results clearly and completely presented?

-Are the figures (Tables, Images) of sufficient quality for clarity?

Reviewer #1: - It is surprising and counterintuitive that the model predicts that water source improvement and reduced open defecation were associated with prevalence of urinary schistosomiasis. The authors should comment on this.

Reviewer #2: All yes.

Reviewer #3: (No Response)

**Conclusions**

-Are the conclusions supported by the data presented?

-Are the limitations of analysis clearly described?

-Do the authors discuss how these data can be helpful to advance our understanding of the topic under study?

-Is public health relevance addressed?

Reviewer #1: - The authors state that their results effectively predict ward-level parasite prevalence, yet no validation of their model was performed. That is, the predicted prevalence in non-surveyed wards wasn't then tested by conducting actual prevalence testing in those wards to confirm if the model's prediction was accurate. Therefore, the statement that the model "effectively predicted" prevalence is an over-ambitious one and does not reflect the actual data (which are speculative).

- It would have been interesting for the authors to have included even a rough estimate of the cost savings that would have been realized had MDA been restricted to the model-predicted higher prevalence wards.

Reviewer #2: All yes.

Reviewer #3: (No Response)

**Editorial and Data Presentation Modifications?**

Reviewer #1: - Abbreviations should be defined in the text, not in a separate table. Also, some abbreviations in the text aren't defined anywhere in the manuscript (e.g., "IU").

- Introduction, lines 111-113: a statement is made that prevalence is increasing, but absolute numbers of cases are given as supporting evidence instead of percentages, which would be more appropriate.

- Introduction, line 131: delete statement about Tanzania having the 2nd highest burden of schistosomiasis as this is repeated from above.

- Methods, line 246: the word "ZIP" appears here inappropriately.

- Results, line 301: should be "intestinal schistosomiasis" instead of just "intestinal".

- Results, line 302: should be "further inland" not "further inlands".

- Results: several of the Table and Figure references appear as "Error! Reference source not found". Recommend removing the hyperlinks.

- Results, lin 336: the second instance of "prevalence" in this sentence should be deleted.

- Figure 5 appears twice in the document.

Reviewer #2: Minor Revision.

Reviewer #3: (No Response)

**Summary and General Comments**

Reviewer #1: (No Response)

Reviewer #2: (No Response)

Reviewer #3: (No Response)

PLOS authors have the option to publish the peer review history of their article (what does this mean?). If published, this will include your full peer review and any attached files.

Reviewer #1: No

Reviewer #2: Yes: David Gurarie

Reviewer #3: Yes: Luc E. Coffeng
---

## [Decision Letter · Decision Letter 1]

13 Nov 2023

Dear Mr Mathewson,

Thank you very much for submitting your revised manuscript "Enabling targeted mass drug administration for schistosomiasis in north-western Tanzania: Exploring the use of geostatistical modeling to inform planning at sub-district level" for consideration at PLOS Neglected Tropical Diseases. As with all papers reviewed by the journal, your manuscript was reviewed by members of the editorial board and by several independent reviewers. The reviewers appreciated the attention to an important topic. There are a number of minor revisions that deserve your attention. Based on the reviews, we are likely to accept this manuscript for publication, providing that you modify the manuscript according to the review recommendations. 

Sincerely,

Philip T. LoVerde

Guest Editor

Eva Clark

Section Editor

Reviewer's Responses to Questions

**Key Review Criteria Required for Acceptance?**

**Methods**

-Are the objectives of the study clearly articulated with a clear testable hypothesis stated?

-Is the study design appropriate to address the stated objectives?

-Is the population clearly described and appropriate for the hypothesis being tested?

-Is the sample size sufficient to ensure adequate power to address the hypothesis being tested?

-Were correct statistical analysis used to support conclusions?

-Are there concerns about ethical or regulatory requirements being met?

Reviewer #1: (No Response)

Reviewer #3: The authors now better explain how the Kato-Katz (KK) was performed. However, from their reply to the comments, it seems they prepared quadruple slides based on a single faecal sample per person, whereas the text in the manuscript suggests that two faecal samples were collected per person and that two slides were prepared per faecal samples ("two sets of duplicate slides"). Please clarify. Also, in their reply to the reviewer comments, the authors clarify that stools were not homogenised (which is important, as they acknowledge) but this is not mentioned in the revised manuscript (with tracked changes).

I repeat my earlier comment about the use of a separate ZIP model and posthoc kriging, and the problem this approach poses for variable selection. In their reply, the authors explain why they use a geospatial approach (which I agree is appropriate but is not the core of my question) and that they intended to "emulate" the established geospatial approach, such as described by Giorgi, Diggle, etc (which I'm familiar with). The problem is that this emulation approach leads to underestimation of standard errors on coefficients in the first step (the ZIP model), which is also used for variable selection. If the authors would instead use the established geospatial approach itself (rather than emulate it in two steps) and base their variable selection on that, they may end up with fewer predictors in the model (because of more convervative and appropriate estimation of standard errors of coefficients).

**Results**

-Does the analysis presented match the analysis plan?

-Are the results clearly and completely presented?

-Are the figures (Tables, Images) of sufficient quality for clarity?

Reviewer #1: (No Response)

Reviewer #3: Figure 6: it is now clear that the left panel is based on a crude district-level aggregation of survey results, while the right panel is based on ward-level model predictions. As one of the main focus points of the paper is ward-level vs. district-level policy, a logical intermediate step in Figure 6 would be to present a middle panel with district-level results based on aggregated model-predictions (instead of the raw data). The authors could even consider adding a fourth panel indicating ward-level results based on raw data, indicating for which wards no inference would be possible as there was not data. This would be a useful illustration for the reader to appreciate which benefits originate from which part of the approach: the use of a geospatial model and the consideration of ward vs. districts as unit of aggregation.

**Conclusions**

-Are the conclusions supported by the data presented?

-Are the limitations of analysis clearly described?

-Do the authors discuss how these data can be helpful to advance our understanding of the topic under study?

-Is public health relevance addressed?

Reviewer #1: The authors removed the word "effectively" from the statement that their model can be used to predict the prevalence of schistosomiasis, but the point of the prior comment was that there has been no validation of the model's prediction (i.e., comparing predicted prevalence against real data). It was the entire statement that is problematic, not just the word "effectively". Please comment.

Reviewer #3: (No Response)

**Editorial and Data Presentation Modifications?**

Reviewer #1: (No Response)

Reviewer #3: (No Response)

**Summary and General Comments**

Reviewer #1: There are several spelling errors in the newly added text (e.g., "Tanzania" and "because") are misspelled.

Reviewer #3: Thank you for carefully addressing my previous comments. Based on the new information, I have only a few points for clarification and improvement.

PLOS authors have the option to publish the peer review history of their article (what does this mean?). If published, this will include your full peer review and any attached files.

Reviewer #1: No

Reviewer #3: Yes: Luc E. Coffeng

Figure Files:

Data Requirements:

Reproducibility:

References

---

## [Decision Letter · Decision Letter 2]

14 Dec 2023

Dear Mr Mathewson,

Thank you very much for submitting your manuscript "Enabling targeted mass drug administration for schistosomiasis in north-western Tanzania: Exploring the use of geostatistical modeling to inform planning at sub-district level" for consideration at PLOS Neglected Tropical Diseases. As with all papers reviewed by the journal, your manuscript was reviewed by members of the editorial board and by several independent reviewers. The reviewers appreciated the attention to an important topic. Based on the reviews, we are likely to accept this manuscript for publication, providing that you modify the manuscript according to the review recommendations. 

Sincerely,

Eva Clark

Section Editor

Reviewer's Responses to Questions

**Key Review Criteria Required for Acceptance?**

**Methods**

-Are the objectives of the study clearly articulated with a clear testable hypothesis stated?

-Is the study design appropriate to address the stated objectives?

-Is the population clearly described and appropriate for the hypothesis being tested?

-Is the sample size sufficient to ensure adequate power to address the hypothesis being tested?

-Were correct statistical analysis used to support conclusions?

-Are there concerns about ethical or regulatory requirements being met?

Reviewer #1: (No Response)

Reviewer #3: I think there might be a typo in the revised description of the KK procedure, where the authors say "homogenous" where I suspect they actually mean "heterogeneous":

“The Kato Katz technique was used for intestinal schistosomiasis, with processed samples prepared on four slides (two sets of duplicate slides) using thick smear for two technicians to evaluate. Thick smears for each of the four slides were prepared using distinct areas of the stool sample to account for a potentially *homogeneous* distribution of eggs within fecal samples.” 

If eggs are distributed homogenously across the stool specimen, it would be irrelevant that the second slide was prepared from a distinctly other area than the first slide.

**Results**

-Does the analysis presented match the analysis plan?

-Are the results clearly and completely presented?

-Are the figures (Tables, Images) of sufficient quality for clarity?

Reviewer #1: (No Response)

Reviewer #3: (No Response)

**Conclusions**

-Are the conclusions supported by the data presented?

-Are the limitations of analysis clearly described?

-Do the authors discuss how these data can be helpful to advance our understanding of the topic under study?

-Is public health relevance addressed?

Reviewer #1: (No Response)

Reviewer #3: (No Response)

**Editorial and Data Presentation Modifications?**

Reviewer #1: (No Response)

Reviewer #3: (No Response)

**Summary and General Comments**

Reviewer #1: The authors have adequately responded to my prior comments and concerns. Just one final minor point that needs to be addressed: in response to Reviewer #3, the authors have added language regarding the KK method to determine fecal egg counts. Instead of making the statement that sampling from 4 separate areas of the fecal sample was done to account for a potentially "homogenous" distribution of eggs, I think that they mean "heterologous".

Reviewer #3: (No Response)

PLOS authors have the option to publish the peer review history of their article (what does this mean?). If published, this will include your full peer review and any attached files.

Reviewer #1: No

Reviewer #3: Yes: Luc E. Coffeng

Figure Files:

Data Requirements:

Reproducibility:

References

---

## [Editor Report · Decision Letter 3]

2 Jan 2024

Dear Mr Mathewson,

We are pleased to inform you that your manuscript 'Enabling targeted mass drug administration for schistosomiasis in north-western Tanzania: Exploring the use of geostatistical modeling to inform planning at sub-district level' has been provisionally accepted for publication in PLOS Neglected Tropical Diseases.

Best regards,

David Joseph Diemert, M.D.

Academic Editor

Eva Clark

Section Editor

---

## [Editor Report · Acceptance letter]

10 Jan 2024

Dear Mr Mathewson,

We are delighted to inform you that your manuscript, "Enabling targeted mass drug administration for schistosomiasis in north-western Tanzania: Exploring the use of geostatistical modeling to inform planning at sub-district level," has been formally accepted for publication in PLOS Neglected Tropical Diseases.

Best regards,

Shaden Kamhawi

co-Editor-in-Chief

Paul Brindley

co-Editor-in-Chief
